# WEBCIR: BRINGING NEW WEB CULTURE CONCEPTS TO COMPOSITIONAL IMAGE RETRIEVAL

## ABSTRACT

This paper proposes an attention-editing-based network knowledge infusion method aimed at enhancing the comprehension and utilization of complex web-scale knowledge in Compositional Image Retrieval (CIR) models. Addressing the limitations of conventional multimodal models in processing massive web knowledge, this study develops an innovative attention-guided knowledge infusion framework through the construction of a structured knowledge-enhanced dataset. The proposed method achieves progressive transmission of web knowledge from coarse to fine granularity via a carefully designed prompt localization system and a hierarchically controlled masking mechanism. Specifically, structured prompt templates encode web knowledge into learnable semantic units, while dynamic attention editing governs the knowledge injection process, enabling the model to adaptively filter and integrate heterogeneous multi-source web knowledge. Experimental results demonstrate that this approach not only significantly improves the model's efficiency in capturing implicit web knowledge but also effectively mitigates knowledge conflicts and redundancy issues. Our work establishes a new technical paradigm for knowledge distillation and transfer in multimodal retrieval systems.

## 1 INTRODUCTION

Compositional Image Retrieval (CIR) is an advanced task in multimodal understanding Baldrati et al. (2023), Jiang et al. (2024), Gu et al. (2024), Zhou et al. (2024b). The primary objective of CIR is to retrieve target images from a large gallery that align with the combined intent of a reference image and a textual modification instruction Zhou et al. (2024a), Song et al. (2025). For example, given an image of a cat and the instruction "turn it into a dog," the model must return an image that preserves the pose, scene, and composition of the original image but replaces the subject with a dog. Unlike traditional image retrieval, CIR requires the model to not only understand the independent semantics of images and text, but also to grasp cross-modal compositional logic and fine-grained semantic transformation relationships. This makes it a critical benchmark for evaluating machine vision-language intelligence Zhang et al. (2025).

Recent advances, driven by large-scale vision-language pretrained models such as Qwen-VL Bai et al. (2023) and Llava Liu et al. (2023), have significantly improved the performance of CIR. The MLLM2Vec Jiang et al. (2025), Zhang et al. (2025), Zhong et al. (2024) paradigm, which integrates visual and textual features through MLLM architectures, has been particularly successful. These models exhibit strong performance in CIR tasks due to their extensive pretraining and well-aligned training strategies Jang et al. (2024), Bai et al. (2025). However, state-of-the-art models still encounter a notable "semantic gap" when dealing with complex, non-literal queries that arise from real-world internet contexts. Existing training datasets, such as COCO, CC3M, and Flickr, predominantly consist of image-text pairs that describe the physical world with clear semantics. In contrast, the real internet world is filled with dynamically evolving, culturally contextualized expressions, including internet memes, slang, and emerging visual styles (e.g., cyberpunk aesthetics). While existing models can easily understand basic image editing commands like "make it black and white" or "show the object in different sizes," they struggle to interpret more abstract instructions such as "make it look like a 90s anime screenshot with a glitch effect" or "give it a dystopian corporate vibe," which require external knowledge and cultural context.

The essence of this gap lies in the model's inability to effectively capture and leverage large-scale, heterogeneous, and dynamically changing internet knowledge. This limitation poses a core challenge that hinders the widespread practical application of CIR technology Zhong et al. (2024). To address these challenges, we propose a novel Attention-Editing-based Network Knowledge Infusion framework, which systematically integrates complex, internet-scale knowledge into CIR models. Specifically, we make the following contributions:

- We introduce a scalable data generation method based on large-model distillation, and construct and release WebCIR, the first CIR dataset focused on internet culture and complex semantic understanding.

- We design a novel attention-editing network framework that infuses external knowledge into the model in a controlled manner, enabling fine-grained alignment of cross-modal features.

- Extensive experiments on multiple public benchmark datasets, including CIRR and FashionIQ, demonstrate that our method significantly outperforms current state-of-the-art models, validating the effectiveness and superiority of our proposed framework.

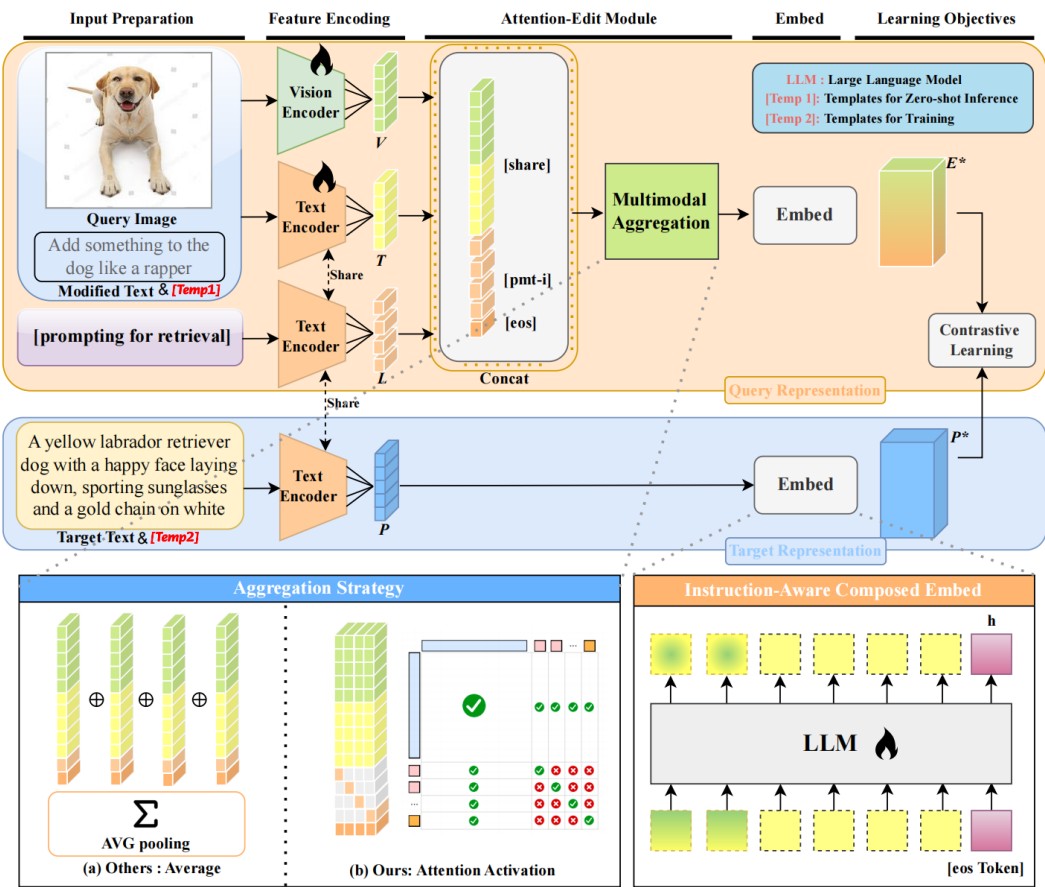

Figure 1: The overall architecture of the proposed framework. The pipeline consists of three main stages: (1) Input Preparation and Encoding: The query image and modification text are processed by shared Vision and Text Encoders to extract unimodal features. (2) Attention-Edit Module: We introduce a novel module that fuses visual and textual cues. Unlike standard aggregation strategies that use simple average pooling , our method employs an Attention Activation mechanism to dynamically weigh features. (3) Learning Objectives: The model produces an Instruction-Aware Composed Embedding ($E^*$) and is optimized via contrastive learning .

## 2 METHOD

### 2.1 BACKGROUND

Composed Image Retrieval stands as a pivotal challenge in cross-modal understanding. Its primary objective is to retrieve a target image from a corpus that satisfies a composite semantic requirement derived from a multimodal query. Specifically, the query combines visual content from a reference image with textual instructions describing semantic modifications. Unlike traditional image-text retrieval, CIR demands that models not only comprehend the independent semantics of visual and linguistic modalities but also master the compositional logic to align visual cues with complex linguistic constraints.

Formally, given a query pair $Q = (I_{ref}, T_{mod})$, where $I_{ref}$ denotes the reference image and $T_{mod}$ represents the modification text, the system aims to retrieve a target image $I_{tgt}$. The retrieved $I_{tgt}$ must semantically satisfy the condition of "being based on $I_{ref}$ and modified according to $T_{mod}$." For instance, providing an image of a cat with the instruction "change to a dog" requires the model to return an image of a dog while preserving the posture, composition, and background of the original reference.

The principal technical challenges inherent to CIR include:

1. **Cross-Modal Semantic Fusion:** The effective integration of visual and textual representations to generate a unified compositional query embedding that accurately encapsulates the modified semantic intent.

2. **Asymmetric Semantic Mapping:** The ability to handle non-linear semantic transformations, such as attribute replacement (e.g., "red → blue"), style transfer (e.g., "oil painting → watercolor"), and object substitution (e.g., "cat → dog").

3. **Fine-Grained Semantic Alignment:** The precise alignment across modalities, particularly when processing abstract, metaphorical, or culturally specific contexts (e.g., internet memes), which necessitates capturing non-literal correspondences between visual and textual elements.

**Evolution and Limitations of Existing Approaches.** Early methodological approaches predominantly relied on late-fusion techniques, utilizing simple feature concatenation or linear transformations. While adequate for straightforward attribute modification, these shallow strategies exhibit limitations in executing complex semantic operations or interpreting non-literal meanings. Recent advancements in pre-trained cross-modal models have established attention-based fusion as the prevailing paradigm. By leveraging cross-attention modules, these approaches facilitate deep interaction between visual and linguistic information, offering innovative solutions for CIR tasks.

Despite these advancements, contemporary methods continue to face critical limitations. First, data scarcity in high-semantic contexts remains a bottleneck; conventional datasets typically feature straightforward factual expressions and lack coverage of the informal, *high-semantic-density* content prevalent in internet culture. Second, suboptimal complex reasoning hinders current models from generalizing effectively when faced with intricate compositional instructions that require multi-step reasoning or abstract semantic mapping.

To address these challenges, this study proposes synergistic enhancements at both the data and model levels. We construct a novel dataset enriched with contemporary internet culture knowledge to bolster non-literal semantic comprehension. Complementing this, we introduce an attention-editing mechanism designed to refine the parsing and execution of intricate instructions. Figure 1 illustrates the complete training and inference pipeline of our proposed framework.

### 2.2 WEBCULTUREAUGMNET: WEB SEMANTIC DATASET DESIGN

#### 2.2.1 CONSTRUCTION OF WEB-CULTURE PERCEPTION DATASET

We introduce the Web-Culture Perception Dataset, designed to systematically encode dynamic internet cultural elements into computable visual-semantic units via large model distillation. Utilizing a Chain-of-Thought strategy with GPT-4o, our pipeline transforms static source descriptions into

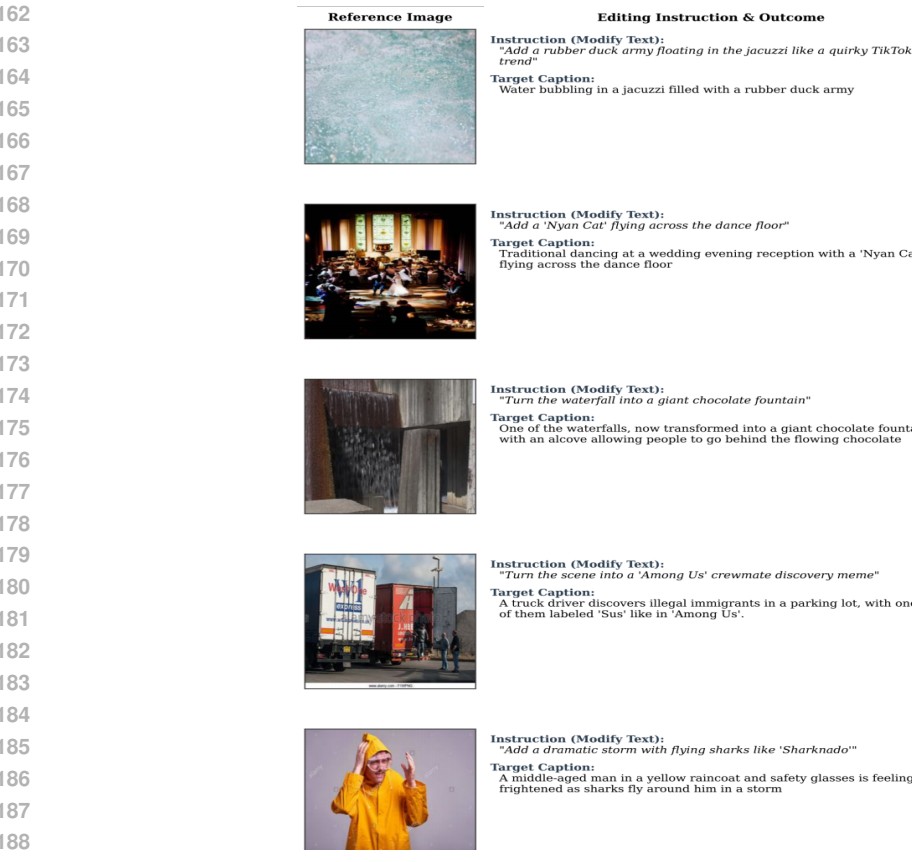

Figure 2: Qualitative examples of the collected triplets. Each sample consists of a reference image, a natural language modification instruction, and a target caption describing the desired visual outcome.

instruction-following triplets that embody "trendy" modifications, ranging from viral meme transformations (e.g., "Doge" style) to subcultural symbol additions like cyberpunk aesthetics. This paradigm surpasses the static limitations of traditional benchmarks (e.g., CIRR) by strictly enforcing internet-native constraints within the generation process. A critical challenge in CIR lies in the prohibitive cost and complexity of acquiring ground-truth modified images ($I_{tgt}$). To address this, we adopt a text-supervised training strategy using the generated triplets ($I_{src}, T_{mod}, T_{desc}$). This approach relies on the cross-modal alignment established during the pre-training phase: since the embedding space aligns visual and textual representations, optimizing the model to retrieve the target text description ($T_{desc}$) effectively serves as a semantic proxy for retrieving the corresponding target image ($I_{tgt}$). This mechanism allows us to bypass the need for explicit visual pairs while ensuring the generated embeddings remain robust for the final image retrieval task.

### 2.2.2 MODEL DISTILLATION-BASED DATA GENERATION PIPELINE

The generation pipeline constructed in this study adopts a GPT-4o-driven CoT framework, achieving precise injection of cultural elements through hierarchical semantic parsing. The technical advantages of this pipeline manifest in three dimensions: First, at the semantic understanding level, the system analyzes not only explicit visual cues in images (e.g., object poses, background composition) but also captures metaphorical expressions unique to web culture (e.g., emotional mapping in memes), enabling generated instructions to transcend traditional simple paradigms like "color replacement/object addition/deletion" and achieve complex cultural adaptations such as "adding influencer filters to portraits". Second, regarding quality control, a three-tier verification mechanism ensures modification rationality, including CLIP cross-modal similarity computation, automatic semantic conflict detection, and manual expert review. Finally, in data organization, all generated 88,694 triplets (source image, culturally adapted instruction, target description) follow standard-

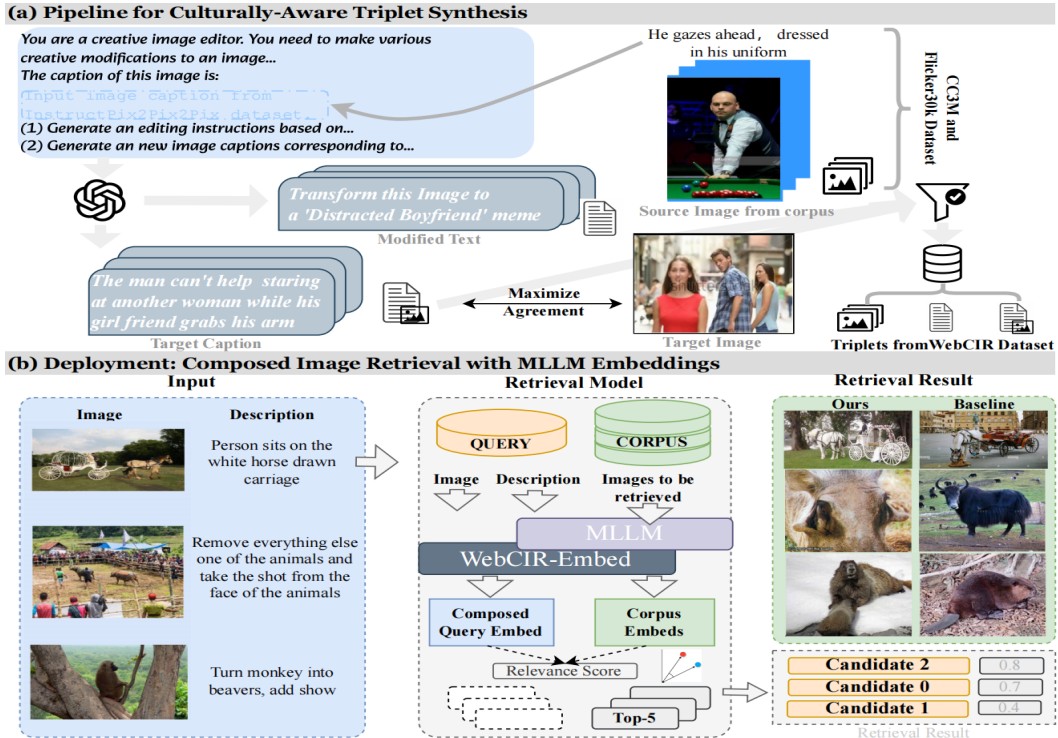

Figure 3: Overview of the WebCIR framework. (a) Data Synthesis Pipeline: We leverage a Large Language Model to generate creative editing instructions and corresponding target captions based on source datasets. This process constructs high-quality triplets for training. (b) Deployment for Composed Image Retrieval: The MLLM processes the composed query (reference image + text) to retrieve the target image from the corpus. The qualitative comparison (bottom) demonstrates that our method accurately captures complex semantic modifications.

ized JSON schema, supporting dual requirements of visual cue preservation and cultural semantic enhancement. Compared to traditional manual annotation methods, this pipeline systematically covers a continuous semantic space from concrete object modification to abstract style transfers (e.g., "cyberpunk style" conversion) through the emergent capabilities of large language models, providing high-quality data foundation for training culture-sensitive compositional image retrieval models. The detailed Template adopted by the pipeline can be found in Appendix A.1 of this paper, and the data overview is shown in Figure 3.

## 2.3 MULTIMODAL EMBEDDING VIA ATTENTION-AWARE EDITING

**Model Architecture.** Our multimodal embedding extraction framework processes heterogeneous inputs through three coordinated components:

1. **Visual Pathway:** For image input $I$, patch embeddings are generated by a Vision Transformer (ViT) and projected into the text embedding space via adapter network $A_v$:

$$\mathbf{E}_v = A_v(\text{ViT}(I)) \in \mathbb{R}^{N_v \times d} \tag{1}$$

2. **Textual Pathway:** Text inputs are tokenized into embeddings:

$$\mathbf{E}_t \in \mathbb{R}^{N_t \times d} \tag{2}$$

The unified representation is constructed through:

$$\mathbf{h} = L\big( \big[ \underbrace{A_v(\text{ViT}(I))}_{\mathbf{E}_v \text{ or } \emptyset} ; \mathbf{E}_t ; \mathbf{e}_{\text{EOS}} \big] \big)_{[-1,:]} \quad \in \mathbb{R}^d \tag{3}$$

The proposed architecture exhibits three key properties: (1) **Modality Fusion** through concatenation $[\,\cdot\,;\,\cdot\,]$ preserves positional encoding while enabling cross-modal attention; (2) **Context Aggregation** via causal transformer $L$ progressively integrates information left-to-right, with the final [EOS] token state $\mathbf{h}$ encapsulating complete context; (3) **Unified Representation** handling both visual-text ($\mathbf{E}_v \neq \emptyset$) and text-only ($\mathbf{E}_v = \emptyset$) inputs within the same framework.

This design ensures that $\mathbf{h}$ always encodes instruction-aware representations, whether processing single or multiple modalities. The causal attention mechanism guarantees that all preceding tokens contribute to the final representation through the transformer's autoregressive property.

**Attention editing.** We propose an innovative semantic alignment framework that combines attention editing with contrastive learning for precise semantic space modeling. The core innovation lies in the hierarchical attention mask design. Given an input sequence $\mathbf{X} = \{\mathbf{x}_1, ..., \mathbf{x}_L\}$ of length $L$, we first instantiate structured templates from a prompt fragment library $\mathcal{P} = \{P_1, ..., P_K\}$, where each fragment $P_k$ is encoded by the tokenizer with recorded position IDs and token IDs.

The attention editing process is defined as:

$$\text{Attention}(\mathbf{Q}, \mathbf{K}, \mathbf{V}, \mathbf{M}) = \text{softmax}\left(\frac{\mathbf{Q}\mathbf{K}^T}{\sqrt{d_k}} \odot \mathbf{M}\right)\mathbf{V} \tag{4}$$

where $\mathbf{Q}$, $\mathbf{K}$, $\mathbf{V}$ represent query, key, and value matrices respectively, $d_k$ is the key dimension, and $\mathbf{M} \in \{0, 1\}^{LL}$ is the hierarchical attention mask matrix with elements satisfying:

$$M_{ij} = \begin{cases} 1 & \text{for intra-template positions } (i, j) \in \mathcal{U}_k \times \mathcal{U}_k \\ 1 & \text{for global prefix tokens } i \leq L_{\text{prefix}} \\ 1 & \text{for special token cross-links} \\ 0 & \text{otherwise} \end{cases} \tag{5}$$

This mask creates block-diagonal attention patterns over semantic units $\{\mathcal{U}_k\}_{k=1}^K$ while preserving global connectivity for [CLS] tokens.

**Triple-Level Representation Transformation.** The semantic enhancement process through attention editing comprises three stages:

1) **Semantic Gating**:

$$\mathbf{h}_i^{\text{gate}} = \sigma\left(\sum_{j=1}^L M_{ij}\mathbf{W}_g\mathbf{x}_j\right) \odot \mathbf{x}_i \tag{6}$$

where $\sigma$ is the sigmoid function and $\mathbf{W}_g$ is a learnable parameter matrix. This operation implements implicit gating within the LLM, enabling automatic learning of semantic unit importance weights.

2) **Context Enhancement**:

$$\mathbf{h}_i^{\text{enh}} = \mathbf{x}_i + \sum_{j=1}^L M_{ij}\alpha_{ij}\mathbf{W}_e[\mathbf{x}_i; \mathbf{x}_j] \tag{7}$$

where $\alpha_{ij}$ denotes attention weights and $\mathbf{W}_e$ is a projection matrix.

3) **Cross-Unit Suppression**:

$$\mathbf{h}_i^{\text{supp}} = \mathbf{x}_i - \sum_{j:M_{ij}=0} \beta_{ij}\mathbf{W}_s\mathbf{x}_j \tag{8}$$

with $\beta_{ij}$ as suppression weights and $\mathbf{W}_s$ as the suppression matrix.

The final representation integrates these components via layer normalization:

$$\mathbf{h}_i = \text{LayerNorm}\left(\mathbf{g} \odot (\mathbf{h}_i^{\text{gate}} + \mathbf{h}_i^{\text{enh}} + \mathbf{h}_i^{\text{supp}})\right) \tag{9}$$

### 2.3.1 CONTRASTIVE LEARNING OBJECTIVE

We construct an improved InfoNCE loss using triplet data $\mathcal{D} = \{(i, t, c_r)\}$:

$$\mathcal{L} = -\log \frac{\phi(\mathbf{h}_{it}, \mathbf{h}_{c_r})}{\phi(\mathbf{h}_{it}, \mathbf{h}_{c_r}) + \sum_{n \in \mathcal{N}} \phi(\mathbf{h}_{it}, \mathbf{h}_n)} \tag{10}$$

where the similarity function $\phi$ is defined as:

$$\phi(\mathbf{u}, \mathbf{v}) = \exp\left(\frac{\mathbf{u}^T \mathbf{v}}{\tau \|\mathbf{u}\| \|\mathbf{v}\|}\right) \tag{11}$$

During training, we employ a dynamic hard negative mining strategy, selecting high-similarity non-target samples from retrieval results to form the negative set $\mathcal{N}$. For inference, zero-shot retrieval is achieved through maximum similarity matching:

$$\hat{c} = \arg\max_{c \in \mathcal{C}} \frac{\mathbf{h}_q^T \mathbf{h}_c}{\|\mathbf{h}_q\| \|\mathbf{h}_c\|} \tag{12}$$

The framework enhances zero-shot generalization through differentiated instruction template designs ($\mathcal{T}_{\text{train}}$ for training and $\mathcal{T}_{\text{test}}$ for inference).

## 3 EXPERIMENTS AND RESULTS

Our experiments are conducted on a computational cluster equipped with four NVIDIA RTX 3090 GPUs, where the proposed framework demonstrates superior computational efficiency compared to both comparable and larger-scale models. For comprehensive evaluation of zero-shot CIR performance, we employ three distinct benchmarks: CIRR Liu et al. (2021) for real-world compositional queries with diverse natural images, FASHIONIQ Wu et al. (2021) for domain-specific fashion attribute retrieval, and our novel POKEMON synthetic dataset generated exclusively from image collections to test out-of-distribution robustness. This evaluation suite collectively assesses model capabilities across conventional, specialized, and synthetic scenarios using recall rate as the primary metric.

We establish rigorous comparisons against three state-of-the-art approaches representing major CIR paradigms: Pic2Word Saito et al. (2023) as the CLIP-based baseline demonstrating strong image-text alignment, MCL Li et al. (2024) as the cutting-edge LLM-based approach employing multi-modal contrastive learning, and the 20-billion parameter GME-Qwen2-VL-2B Zhang et al. (2025) vision-language foundation model providing scale comparison. All models utilize ViT-L (224×224) visual backbones to ensure controlled experimental conditions, with this selection strategy serving to cover dominant research directions while highlighting our method's efficiency advantages against massive architectures.

Standardized evaluation protocols govern all experiments to ensure reproducibility, with particular attention to fair comparison across different model scales and architectures. The POKEMON dataset construction and detailed efficiency analysis are provided in the Appendix A.2, complementing the main experimental results that demonstrate our framework's consistent improvements in both accuracy and computational efficiency across all evaluation scenarios.

### 3.1 EXPERIMENTAL RESULTS

Through systematic evaluation of Zero-Shot CIR models, this study compares the performance of four advanced models on three representative benchmark datasets (M-BEIR:CIRR, M-BEIR:FASHIONIQ, and Benchmark:POKEMON). The experimental results demonstrate that our proposed WebCIR model shows significant advantages on most evaluation metrics, particularly excelling in more challenging cross-modal retrieval tasks.

On M-BEIR:CIRR, WebCIR achieves 24.68% Recall@1, significantly outperforming MCL (15.24%) and GME (14.85%). The advantage persists with 79.76% Recall@50 (3-4% higher than baselines) and an 11.6% improvement in the comprehensive metric (2.8298), demonstrating robust cross-modal understanding likely from web-scale pretraining.

Table 1: Comparison of Zero-Shot CIR Models on Validation Sets

| Model | Recall@K | | | | | Metric |
|---|---|---|---|---|---|---|
| | @1 | @5 | @10 | @25 | @50 | |
| **M-BEIR:CIRR** | | | | | | |
| Pic2word | 0.1086 | 0.2404 | 0.4406 | 0.5833 | 0.6574 | 2.0303 |
| MCL | 0.1524 | 0.3942 | 0.5503 | 0.6782 | 0.7608 | 2.5359 |
| GME | 0.1485 | 0.4173 | 0.5380 | 0.6691 | 0.7594 | 2.5323 |
| **Ours: WebCIR** | 0.2468 | 0.4985 | 0.5782 | 0.7087 | 0.7976 | 2.8298 |
| **M-BEIR:FASHIONIQ** | | | | | | |
| Pic2word | 0.0827 | 0.2076 | 0.2812 | 0.3664 | 0.4314 | 1.3693 |
| MCL | 0.1004 | 0.2538 | 0.3215 | 0.3887 | 0.4457 | 1.5101 |
| GME | 0.0783 | 0.1894 | 0.2577 | 0.3486 | 0.4282 | 1.3022 |
| **Ours: WebCIR** | 0.1008 | 0.2097 | 0.2704 | 0.3648 | 0.4403 | 1.3860 |
| **Benchmark:POKEMON** | | | | | | |
| Pic2word | 0.0529 | 0.1518 | 0.2545 | 0.3596 | 0.4529 | 1.2717 |
| MCL | 0.0807 | 0.2022 | 0.2888 | 0.4067 | 0.5180 | 1.4964 |
| GME | 0.0274 | 0.1305 | 0.2034 | 0.3172 | 0.4259 | 1.1044 |
| **Ours: WebCIR** | 0.0783 | 0.2122 | 0.3087 | 0.4438 | 0.5507 | 1.5937 |

Note: The Metric value is the sum of all Recall@K values.

Figure 4: Recall@K performance comparison on three zero-shot composed image retrieval benchmarks. We compare the proposed WebCIR against state-of-the-art baselines (Pic2word, MCL, and GME) on the CIRR, FashionIQ, and Pokemon datasets.

For M-BEIR:FASHIONIQ, all models show comparable performance due to fashion's inherent ambiguity. WebCIR slightly leads in Recall@1 (10.08% vs. 10.04%) but trails in Recall@5 (20.97% vs. 25.38%), suggesting challenges with subjective fashion semantics.

Most notably on Benchmark:POKEMON, WebCIR shows strongest generalization with 55.07% Recall@50, significantly leading MCL (51.80%) and Pic2word (45.29%). GME's exceptionally low Recall@1 (2.74%) reveals architecture sensitivity to data distributions.

Computationally, WebCIR maintains leading Metric scores across all datasets, particularly showing 6.5% improvement on POKEMON, with stable high-recall performance indicating effective precision-coverage balance.

Despite the promising results, this study presents certain limitations. First, the performance bottleneck observed on the FashionIQ dataset suggests a need for more specialized modules to capture fine-grained fashion semantics. Second, while WebCIR demonstrates robust overall perfor-

Table 2: Analysis of Ablation Study Results

| Setting | CIRR | | | FASHIONIQ | | | POKEMON | | | Avg. | | |
|---|---|---|---|---|---|---|---|---|---|---|---|---|
| | @1 | @5 | @10 | @1 | @5 | @10 | @1 | @5 | @10 | @1 | @5 | @10 |
| **Fine-tuning strategy** | | | | | | | | | | | | |
| Lora r=4 | 23.3 | 46.2 | 56.7 | 7.4 | 16.3 | 21.8 | 7.2 | 20.7 | 30.3 | 12.6 | 27.7 | 36.3 |
| Lora r=8 | 23.3 | 46.2 | 56.7 | 7.4 | 16.3 | 21.8 | 7.2 | 20.7 | 30.4 | 12.6 | 27.7 | 36.3 |
| Lora r=32 | 23.1 | 45.9 | 56.6 | 7.3 | 16.2 | 21.8 | 7.2 | 20.7 | 30.3 | 12.5 | 27.6 | 36.2 |
| **Training data organization** | | | | | | | | | | | | |
| w/o hard-negative | 12.8 | 38.2 | 48.7 | 5.3 | 10.8 | 13.2 | 7.8 | 19.4 | 26.6 | 8.6 | 22.8 | 29.5 |
| **Modeling** | | | | | | | | | | | | |
| w/o attention-edit | 16.8 | 33.6 | 42.3 | 4.3 | 9.3 | 12.0 | 5.9 | 18.2 | 26.3 | 9.0 | 20.4 | 26.9 |
| **Dataset** | | | | | | | | | | | | |
| sub dataset (5k) | 23.0 | 45.8 | 55.5 | 7.3 | 16.1 | 21.7 | 6.9 | 20.7 | 30.4 | 12.4 | 27.5 | 35.9 |
| sub dataset (25k) | 23.0 | 45.9 | 56.1 | 7.4 | 16.3 | 22.1 | 6.4 | 18.6 | 28.9 | 12.3 | 26.9 | 35.7 |
| Flickr-32k (32k) | 23.8 | 48.0 | 58.1 | 9.1 | 20.0 | 26.1 | 5.7 | 19.4 | 27.3 | 12.9 | 29.1 | 37.2 |
| **Prompt Modification** | | | | | | | | | | | | |
| RUN 1 (prompt_1) | 22.0 | 44.0 | 54.0 | 6.4 | 14.6 | 19.6 | 6.6 | 19.7 | 27.9 | 11.7 | 26.1 | 33.8 |
| RUN 2 (prompt_2) | 21.7 | 44.0 | 53.7 | 6.6 | 15.0 | 20.0 | 6.7 | 19.5 | 28.2 | 11.7 | 26.2 | 34.0 |
| prompt (num = 6) | 21.2 | 43.6 | 52.9 | 6.5 | 14.8 | 19.9 | 6.2 | 18.5 | 27.3 | 11.3 | 25.6 | 33.4 |

Note: Detailed training configurations are documented in the Appendix.

mance—as qualitatively evidenced by the retrieval visualizations in Appendix A.4—fluctuations in certain intermediate metrics indicate that the model's adaptability to diverse query types requires further refinement. These findings illuminate future research directions, specifically the exploration of domain-adaptive pre-training strategies and advanced cross-modal alignment mechanisms. In conclusion, our experimental results validate the efficacy of WebCIR, establishing a solid baseline and providing new technical pathways for zero-shot CIR research.

# 4 ABLATION STUDY

Through systematic analysis of the ablation study results (Table 2), we draw the following key findings. Regarding fine-tuning strategies, LoRA methods with different rank parameters demonstrate similar performance, where settings with r=4 and r=8 achieve identical accuracy across all datasets, while r=32 shows only marginal degradation (average @1 drops by 0.1%), indicating the model's robustness to LoRA rank selection.

Notably, removing hard-negative sampling leads to significant performance degradation, particularly on CIRR dataset with a 10.5 percentage point drop in @1 and 2.1 points on FASHIONIQ, demonstrating the crucial role of hard negatives in learning discriminative features. For modeling approaches, removing the attention-edit module causes comprehensive performance decline across all datasets, with CIRR @1 decreasing by 6.5 points and FASHIONIQ by 3.1 points, validating the module's importance in feature editing. Analysis of training data scale reveals that using the complete Flickr-32k dataset yields better performance compared to CC3M subsets, especially on FASHIONIQ with 1.8 points @1 improvement, suggesting larger datasets enhance model generalization. Prompt modification experiments show relatively minor differences, with performance variations within reasonable ranges. Overall, hard-negative sampling and attention-edit module emerge as critical factors affecting model performance, while LoRA rank selection and data scale exhibit more stable influence patterns.

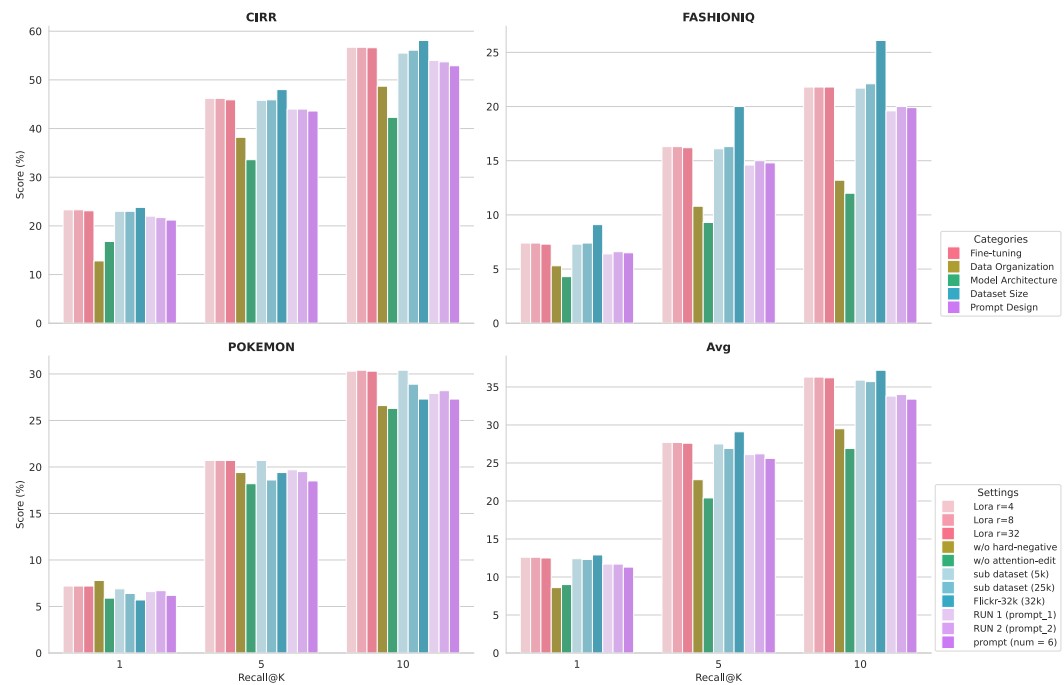

Figure 5: Ablation study and hyperparameter analysis. We investigate the impact of different design choices on retrieval performance across CIRR, FashionIQ, and Pokemon datasets. The analysis covers five categories: fine-tuning settings (LoRA rank), data organization (hard-negatives), model architecture (attention-edit), dataset scaling, and prompt design strategies. The "Avg" plot summarizes the overall trend.

## 5 CONCLUSION

This framework presents a novel attention-editing framework for knowledge infusion in CIR systems. The proposed method establishes a principled approach to web-scale knowledge acquisition through structured prompt encoding and dynamic attention modulation, effectively addressing the fundamental challenges of knowledge heterogeneity and semantic alignment in multimodal learning. The framework demonstrates significant potential for advancing knowledge-enhanced vision-language models while maintaining rigorous semantic consistency. These contributions open new research directions for developing more sophisticated knowledge integration paradigms in artificial intelligence systems.

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

## A  APPENDIX

### A.1  DATA DISTILLATION PIPELINE PROMPT TEMPLATES

The distillation process employs two core prompt templates designed to generate high-quality training data for CIR. These templates guide the model to produce semantically controlled modifications, balancing explicit semantic transformations and internet-culture-aware adaptations.

1. Basic Semantic Transformation Template Objective: Ensure precise, atomic-level modifications in generated image pairs. Template: "I am creating a multi-modal dataset for CIR. The goal is to generate pairs of source and target images, along with a modification instruction that performs a single, well-defined transformation (e.g., object color shift, scene replacement, or action adjustment). The instruction should be concise, and the modified description must reflect only the specified change."

Key Requirements: Unidirectional edits: Each instruction induces exactly one salient change (e.g., "Change the dog's fur from brown to white"). Preservation of core semantics: Non-target attributes remain unaltered.

2. Internet-Culture-Augmented Template Objective: Inject internet-meme aesthetics while retaining executable clarity. Template: "I am building a multi-modal dataset for CIR with internet culture elements. Your task is to reference visual details from the source image and craft modification

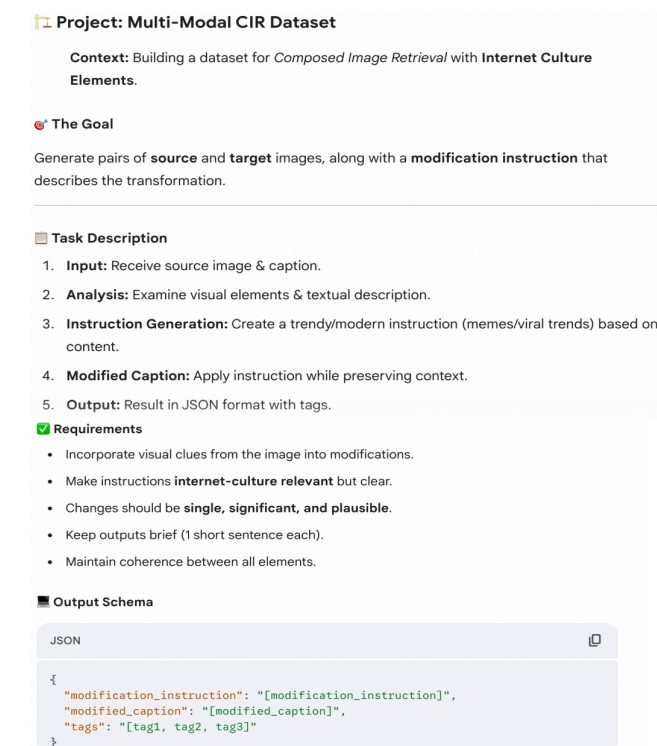

Figure 6: Full Version Prompt Template

instructions that resonate with trending online tropes (e.g., memes, viral expressions, or pop-culture references). Ensure the instruction remains unambiguous and feasible for retrieval."

Key Requirements: Context-aware humor: Modifications should naturally align with the image's content (e.g., turning a neutral cat into a "disturbed loaf cat" meme). Balanced creativity and precision: Avoid overfitting to niche contexts; prioritize widely recognizable cues.

The full version prompt is shown in fig 6

### A.2 POKEMON Benchmark Construction for Online CIR Demonstration

The POKEMON benchmark was developed with a dual purpose: to rigorously assess zero-shot generalization in non-realistic domains and, crucially, to demonstrate the practical utility of our proposed framework in a real-world Online CIR Task scenario. This setup is designed to mirror applications requiring instantaneous retrieval from a large-scale corpus based on compositional queries.

The operational flow of this benchmark utilizes our proposed WebCIR model—powered by the novel *Attention-Editing Mechanism* for robust multimodal representation learning—to bridge offline indexing with online querying, as illustrated in the system architecture diagram (see Figure 8). The process consists of two phases:

- **Offline Corpus Indexing:** We utilize the comprehensive POKEMON image collection from HuggingFace to construct the retrieval corpus. The WebCIR model processes each image in the corpus, leveraging its attention-editing mechanism to generate highly discriminative *Content Embeddings*. These embeddings are stored in a vectorized *Embedding Table* (Emb. Table), ready for efficient approximate nearest neighbor (ANN) search.

- **Online Real-time Retrieval:** During the online phase, a user provides a compositional query (a reference image and modification text). The WebCIR model instantly processes this query into a *Query Embedding*. An ANN search is then executed against the pre-

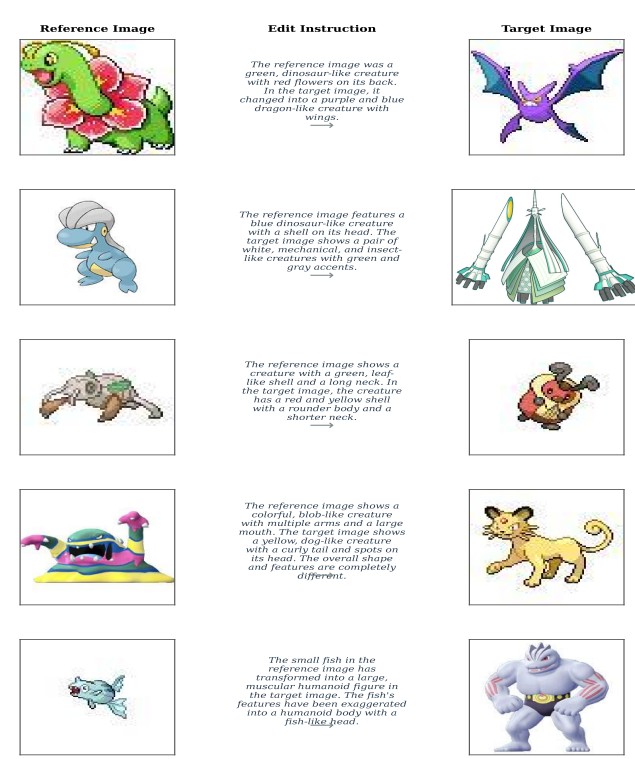

Figure 7: Sample triplets from the Pokemon benchmark. Each example consists of a reference image, a descriptive modification instruction, and the corresponding ground-truth target image.

computed Embedding Table to recall and return the top-K most relevant target images to the user in real-time.

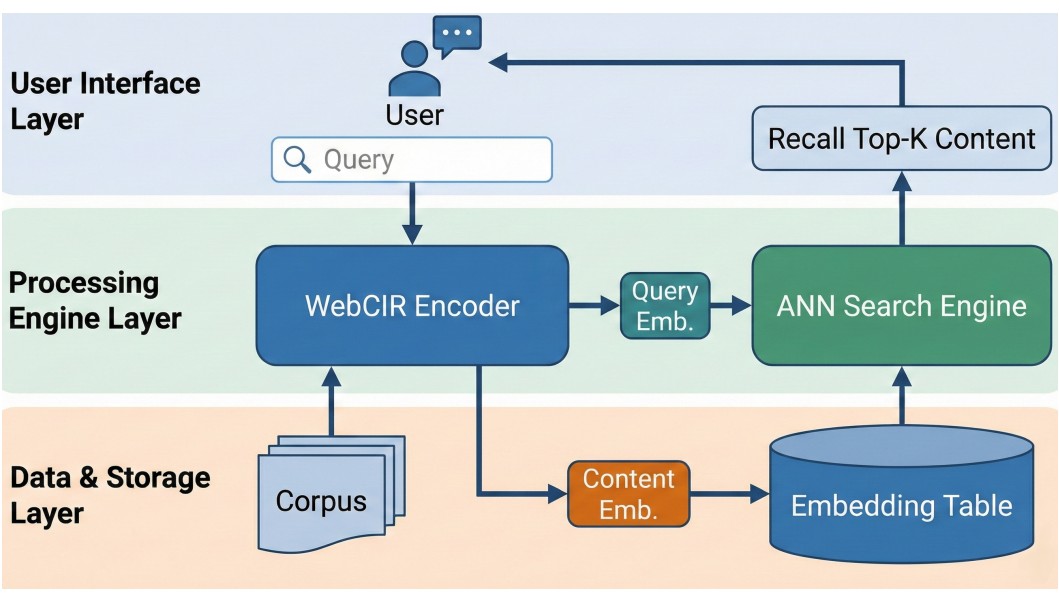

Figure 8: Online Real-time Retrieval

To ensure evaluation fairness and isolate the contribution of our architecture from potential biases in our main training pipeline (which used GPT-4o), the evaluation triplets (query caption, modified text,

and target image) for this benchmark were reconstructed using fundamentally different protocols, specifically utilizing distinct prompt templates and the Qwen2-VL-72B-Instruct model.

## A.3 LLM-Assisted Data Analysis and Visualization

In the data analysis and experiment settings, we employed LLMs as auxiliary tools for expression refinement while maintaining original authorship of all content. The models provided supplementary language suggestions, but all analytical insights, interpretations, and final textual formulations were independently developed and verified by us.

## A.4 Retrieval Result

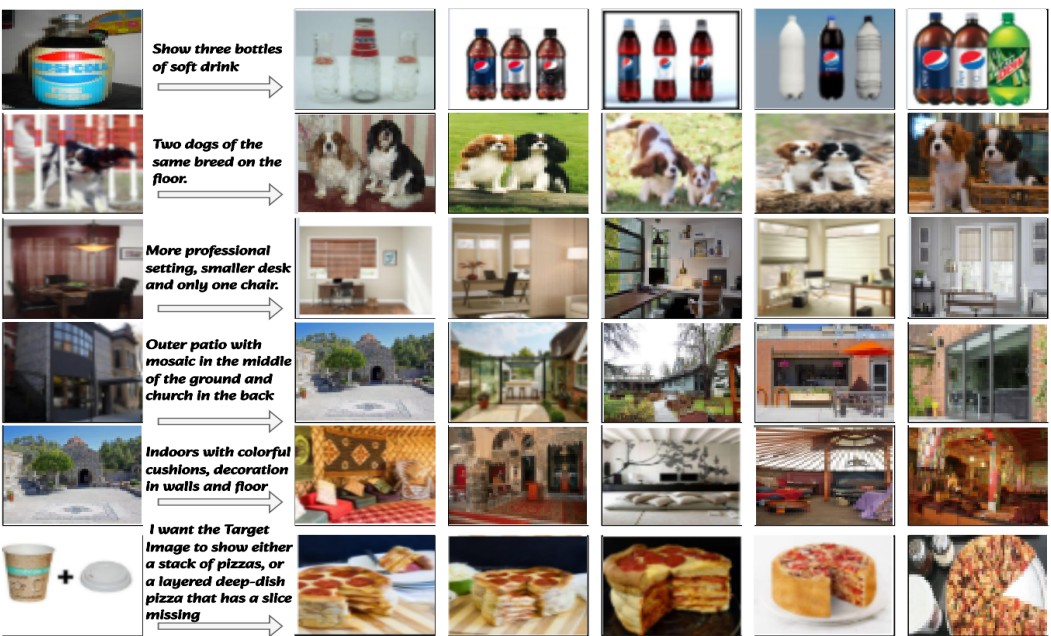

Figure 9: Zero-Shot Retrieval Examples on Real-World Queries

