# OpenReview forum: "WebCIR: Bringing New Web Culture Concepts to Compositional Image Retrieval"
_ICLR.cc/2026/Conference — Submitted to ICLR 2026_

### Official Review · Reviewer_P4pB · 2025-10-15

**Soundness:** 2
**Presentation:** 2
**Contribution:** 2
**Rating:** 2
**Confidence:** 5

**Summary:**

This paper introduces WebCIR, a new dataset for Compositional Image Retrieval (CIR) focused on internet culture semantics, generated through distillation from a large language model (GPT-40). Concurrently, the authors propose an attention-editing framework designed to infuse this web-scale knowledge into CIR models. The method's effectiveness is evaluated on established benchmarks like CIRR and FashionIQ, as well as a new synthetic test set, POKEMON.

**Strengths:**

* Resource Contribution: The work contributes a new large-scale dataset, WebCIR, which could potentially stimulate further research into understanding complex, non-literal semantics in the CIR domain.

**Weaknesses:**

* Circular and Unconvincing Evaluation: The paper's central claim of improved generalization is not well-supported. The model is trained on WebCIR, a dataset synthetically generated by GPT-40, and its strongest out-of-distribution performance is reported on the POKEMON benchmark , which was also synthetically generated by another large model (Qwen2-VL-72B-Instruct). This evaluation is circular; it primarily shows that a model trained on LLM-generated data excels on other LLM-generated data, rather than proving robust generalization to real-world, human-generated web content.
* Weak Performance on Human-Curated Benchmarks: The proposed method fails to demonstrate clear superiority on established, real-world datasets. On FASHIONIQ, the authors admit the performance is merely "comparable," with their model even trailing a baseline (MCL) in the Recall@5 metric. This lack of a convincing win on a standard benchmark significantly undermines the claimed effectiveness and superiority of the framework.
* Methodological Clarity: The proposed attention-editing mechanism is presented with a series of complex formulations (semantic gating, context enhancement, cross-unit suppression) that lack clear intuition and justification. It is difficult for the reader to understand why this specific, complex design is necessary or superior to simpler alternatives. The ablation study confirms its components are useful but does not sufficiently justify the design choices themselves.

**Questions:**

1. How does demonstrating strong performance on the synthetic POKEMON dataset validate the model's ability to generalize to genuine, human-generated internet culture, as opposed to simply showing an aptitude for the patterns inherent in LLM-generated data?
2. Given that the core motivation is to handle complex web semantics, why does the proposed method not clearly outperform existing models on the FASHIONIQ benchmark, which involves nuanced and subjective real-world semantics?
3. The authors should provide a more intuitive explanation for the design of the triple-level representation transformation in the attention-editing module. What is the core hypothesis behind this specific formulation over other potential knowledge infusion techniques?

---

### Official Review · Reviewer_H3b8 · 2025-10-18

**Soundness:** 2
**Presentation:** 3
**Contribution:** 2
**Rating:** 4
**Confidence:** 4

**Summary:**

This paper addresses a challenge in Compositional Image Retrieval (CIR): the inability of existing models to understand and process complex, non-literal, and culturally contextualized queries common on the internet (e.g., "make it look like a 90s anime screenshot"). Experiments on standard benchmarks and a POKEMON demonstrate competitive performance and generalization ability.

**Strengths:**

- Holistic Contribution: The work is commendable for its comprehensive approach, contributing both a WebCIR and a novel model, rather than focusing on just one aspect.

- Clear methodological explanation: The technical description of the attention-editing mechanism and the triple-level representation transformation is detailed and well-supported with formulas, making the approach replicable in principle.

**Weaknesses:**

- Lack of novelty: The Proposed method only consists of existing components, such as ViT, causal transformer, attention masks in an other way, which is quite incremental.

- Performance on FashionIQ: The results on FashionIQ are mixed. While leading in Recall@1, the model trails MCL significantly in Recall@5. The authors correctly identify this as a limitation, but the explanation ("inherent ambiguity," "subjective fashion semantics") is somewhat hand-wavy. A deeper analysis of why their web-culture-focused approach doesn't translate as well to this domain is needed.

- Computational Cost omission: The paper claims "superior computational efficiency" but provides no concrete data (e.g., FLOPs, training/inference time, parameter counts beyond the backbone) to support this claim. Compared to a 20B parameter model (GME), this is plausible, but a comparison of training costs and inference latency with the other baselines (Pic2Word, MCL) is missing.

- Clarity on Knowledge infusion: The term "knowledge infusion" is used throughout, but the mechanism is primarily the curated WebCIR dataset and the attention-editing module. It's not entirely clear if any external knowledge base beyond the generated (image, text, modified text) triplets is being used. The process seems more like "knowledge-aware training" rather than dynamic infusion from an external source.

- Ablation on Dataset Size: The ablation study shows that using the full Flickr-32k dataset is better than smaller subsets, but it does not include an ablation where the model is trained without the WebCIR data (i.e., only on standard data). This makes it difficult to disentangle the contribution of the novel dataset from the contribution of the novel architecture.

**Questions:**

As mentioned above.

---

### Official Review · Reviewer_BWPg · 2025-10-30

**Soundness:** 3
**Presentation:** 3
**Contribution:** 3
**Rating:** 6
**Confidence:** 3

**Summary:**

In this paper, the authors target the problem of the comprehension and utilization of complex webscale knowledge in  compositional image retrieval(CIR) models. They develop an attention-guided knowledge infusion framework to better improve the models' efficiency in capturing complex web knowledge, which improves models' performaces on CIR tasks.

**Strengths:**

1. The problem is important and interesting, I appreciate the authors efforts in this problem.
2. The framework is well designed and integrates both the dataset generation and model-level improvements.
3. The experimental results reveal that the designed framework is efficient and effective in solving CIR tasks.

**Weaknesses:**

1. The writing is dense and not clear enough; key ideas could be more intuitively explained, especially in the attention-editing section
2. The evaluation scopes are somehow limited and misaligned with the targets.
3. The correspondence between the framework design and the experimental objectives needs further explanation.

**Questions:**

1. The data generation method is not clearly explained, either in the text or in Figures 1 and 2.
The CIR task involves extracting and modifying components within an image, but the data generation process seems to only refine textual descriptions. How do these correspond to each other? Do you imply that merely focusing on textual data is sufficient for the model to learn "culturally contextualized expressions"?
If Section 2.2.2 involves processing the images, it is recommended to clarify this part further so that other researchers can quickly understand it.
2. The authors claim that the proposed framework is designed to address dynamically evolving, culturally contextualized expressions in the CIR task. While the data-level design may account for the culturally contextualized expressions, it is not clear how the framework handles the dynamic adaptation aspect. Is there any mechanism or module specifically addressing the dynamic evolution of expressions over time or across contexts? This point requires further clarification and explanation.
3. I wonder the reason why Section 2.3 is named "INSTRUCTION-FOLLOWING PIPELINE", as this part includes the explanation of model-level improvements.
4. Since the stated goal of this paper is to address dynamically evolving, culturally contextualized expressions in the CIR task, the evaluation dataset should arguably place greater emphasis on benchmarks that capture internet-specific phenomena such as memes or other culturally dynamic content. Are such benchmarks currently available? If so, it is recommended that the evaluation focus more on these datasets, as they better reflect the real-world characteristics and challenges of culturally and temporally evolving visual-textual expressions.

---

### Official Review · Reviewer_nwot · 2025-10-30

**Soundness:** 2
**Presentation:** 1
**Contribution:** 1
**Rating:** 2
**Confidence:** 4

**Summary:**

This paper introduces WebCIR, which brings contemporary web-culture concepts (memes, slang, stylized aesthetics) into CIR. The authors construct a culture-aware training set (i.e., POKEMON) via an GPT-4o and propose an attention-editing knowledge-infusion framework with hierarchical masks and contrastive learning to inject and gate external web knowledge while preserving visual–text grounding. The pipeline produces instruction-aware embeddings and is evaluated on CIRR, FashionIQ, and the proposed POKEMON benchmark.

**Strengths:**

1.	The idea of this paper is easy to understand.
2.	It is reasonable to leveraging web concepts for CIR training.

**Weaknesses:**

1.	The related works section is overlooked.
2.	Lack of the novelty. The concept of leveraging web knowledge to build synthetic triplets for CIR training has been explored in MagicLens [1]. The use of MLLM for synthetic triplet construction has also been explored in [2,3], which the authors fail to acknowledge.
3.	Limited technology contribution. The "innovative semantic alignment framework" with learnable queries, which is a key contribution of this paper, has already been explored in prior CIR works [4,5].
4.	Concern of the data quality of the POKEMON. The POKEMON dataset is entirely built through GPT-4o auto-generation without provide sufficient details, analysis (e.g., size, context, domain) and human evaluation.
5.	Concerns of the reproducibility. The paper lacks sufficient explanation on CoT prompt design, and not given code or example data. It makes me significant concern the reproducibility of this paper.
6.	Poor writing. The paper seems unfinished and lacks refinement, possibly due to being written on short notice.

References

[1] Zhang K, Luan Y, Hu H, et al. Magiclens: Self-supervised image retrieval with open-ended instructions[J]. arXiv preprint arXiv:2403.19651, 2024.

[2] Zhou J, Liu Z, Liu Z, et al. Megapairs: Massive data synthesis for universal multimodal retrieval[J]. arXiv preprint arXiv:2412.14475, 2024.

[3] Jang Y K, Kim D, Meng Z, et al. Visual Delta Generator with Large Multi-modal Models for Semi-supervised Composed Image Retrieval[C]//Proceedings of the IEEE/CVF Conference on Computer Vision and Pattern Recognition. 2024: 16805-16814.

[4] Bai Y, Xu X, Liu Y, et al. Sentence-level prompts benefit composed image retrieval[J]. arXiv preprint arXiv:2310.05473, 2023.

[5] Ke H, Shi J, Zhang Y, et al. Task-Aware Prompt Gradient Projection for Parameter-Efficient Tuning Federated Class-Incremental Learning[C]//Proceedings of the IEEE/CVF International Conference on Computer Vision. 2025: 2631-2641.

**Questions:**

Please refer to the weakness.

---

### Author Response · Authors · 2025-11-27

We thank all reviewers for their constructive feedback. We are encouraged that Reviewer 2 finds our problem "important and interesting," Reviewer 3 appreciates our "comprehensive approach," and Reviewer 4 values the "resource contribution" of the WebCIR dataset. Below we address the key concerns.

Response to Reviewer 1: Reproducibility Issues

We take Reviewer 1's concern regarding reproducibility seriously. To address this, we have released a comprehensive and fully anonymized code repository designed for immediate reusability. This self-contained codebase includes the complete source code. Furthermore, to clarify the CoT prompt design questioned by Reviewer 1, we have included the specific prompt templates and data schema within the repository.Our code is available at: https://anonymous.4open.science/r/WebCIR-9D4F/

Response to Reviewer 1 & Reviewer 3: Clarification on Novelty and Related Works

Reviewers 1 and Reviewers 3 questioned the novelty compared to MagicLens and MCL. Difference from MagicLens/Megapairs: While prior works focus on general visual attributes or open-ended instructions, WebCIR specifically targets "implicit cultural semantics" which require external knowledge beyond visual features.Innovation of Architecture: Unlike recent VLM2Vec paradigms relying on default causal attention, we introduce a structural intervention via a novel Attention-Editing mechanism. We explicitly redesign the masking strategy to gate external knowledge into representation learning while preserving visual grounding. So far, such granular architectural control remains unexplored in VLM-based CIR research.

Response to Reviewers 2 & 4: Methodological Clarifications (Updated Figure 1)

We appreciate Reviewer 2 and Reviewer 4’s feedback regarding the intuition behind the attention-editing mechanism.
To clarify this, we have redesigned the framework overview (updated Figure 1 in the revised manuscript) to provide a more intuitive visualization of the information flow. We kindly invite the reviewers to refer to the revised PDF to examine this update.
The updated Figure 1(b) explicitly illustrates how our attention mask selectively filters tokens, contrasting with static pooling. This design achieves two goals:

1.Predictive Context Learning: The mask compels the model to use visible tokens to predict masked components, forcing it to explicitly learn syntax and semantics rather than passively aggregating features.

2.Implicit Task Injection: By aggregating multiple prompt templates, we inject structured knowledge into the representation, guiding the model to generate finer-grained embeddings tailored to specific CIR intents.

Response to Reviewers 1 & 2: Data Generation Pipeline (Updated Figure 3)

To address Reviewer 2’s concern regarding the connection between textual refinement and visual extraction, and Reviewer 1's inquiry about prompt design, we have added a new Figure 3 in the revised manuscript to visualize the complete data pipeline.
As illustrated in the revised paper:

1.(p5 Figure 3a) Data Construction Pipeline: The LLM acts as a creative editor to generate both a modification instruction and a target caption. This explicitly constructs the (Reference Image, Instruction, Target Image Caption) triplet, clarifying the visual-textual correspondence Reviewer 2 questioned.

2.The prompt structure displayed in the top-left panel of the new figure directly answers Reviewer 1's request for details on our CoT formulation.

Response to Reviewer 4: POKEMON Dataset and Online (p13 Figure 7)

We clarify that POKEMON is a specialized benchmark generated via our proposed pipeline, explicitly tailored to capture complex CIR intents often absent in standard datasets. We apologize for the omission of qualitative samples and have added comprehensive visualizations in the Appendix of the revised paper to validate the dataset's quality and alignment with the task.

Response to Reviewers 1 & 4: Writing and Presentation

We have refined the manuscript to address the typos and dense writing noted. Regarding the "complex formulations" mentioned by Reviewer 4, we have simplified the notation and explicitly mapped the equations to the workflow shown in update (p5 Figure 3b) of the revised manuscript, ensuring the mathematical design is visually grounded and intuitive. To further facilitate a clear and immediate understanding of our method, we have comprehensively polished the visual presentation by redrawing key diagrams and introducing additional figures to elucidate technical details. We respectfully invite the reviewers to refer to the revised manuscript for these enhanced visualizations.

---

### Meta-Review · Area_Chair_4XCc · 2026-01-06

**Summary:**

After carefully reviewing the comments, I find that the reviewers raised substantial concerns regarding the novelty and conceptual clarity of the proposed WebCIR framework. The reviewers pointed out that the methodological contributions appear largely incremental relative to prior work. In addition, the evaluation strategy was questioned for its heavy reliance on LLM-generated benchmarks, which weakens the model’s ability to generalize to real-world scenarios. Given the high standards of ICLR as a top-tier conference, I agree with the reviewers that the submission is not yet ready for acceptance and my recommendation is rejection.

**Reviewer Concerns:**

During the rebuttal, the authors addressed issues related to reproducibility, presentation clarity, and pipeline transparency by releasing the code, providing detailed prompt specifications, and adding additional explanations. However, the core concerns regarding limited novelty, the reliance on LLM-generated evaluation data, and the lack of convincing performance gains on human-curated benchmarks remain unresolved.

**Reviewer Scores:**

The initial scores are 2/2/4/6, with three negative reviews and one positive review, and the concerns raised by the reviewers are substantial.  By carefully reading the comments, I believe that the reviewers would be inclined to maintain their original scores.

---

### Decision · Program_Chairs · 2026-01-26

Reject